# Destruction of ERP responses to deviance in an auditory oddball paradigm in amyloid infusion mice with memory deficits

**Bowon Kim[1,2], Jisu Shin[3,4], YoungSoo Kim**![ORCID][3,4,5]*****, **Jee Hyun Choi[1,2]***

**1** Center for Neuroscience, Korea Institute of Science and Technology, Seoul, Republic of Korea, **2** Division of Bio-Medical Science & Technology, KIST School, Korea University of Science and Technology, Seoul, Republic of Korea, **3** Department of Pharmacy, Yonsei University, Incheon, Republic of Korea, **4** Yonsei Institute of Pharmaceutical Science, Yonsei University, Incheon, Republic of Korea, **5** Integrated Science and Engineering Division, Yonsei University, Incheon, Republic of Korea

* y.kim@yonsei.ac.kr (YK); jeechoi@kist.re.kr (JHC)

**Data Availability Statement:** All relevant data are within the manuscript and its Supporting Information files.

**Funding:** This research was supported by the National Research Foundation (NRF) of Korea grant

## Abstract

The amyloid-β (Aβ) oligomer is considered one of the major pathogens responsible for neuronal and synaptic loss in Alzheimer's disease (AD) brains. Although the neurotoxic mechanisms of Aβ have been widely investigated, experimental evidence for the direct linkage between neural signaling and cognitive impairments in association with peptide oligomers is lacking. Here, we conducted an auditory oddball paradigm utilizing an Aβ-infused Alzheimer's disease mouse model and interpreted the results based on Y-maze behavioral tests. We acutely injected Aβ oligomers into the intracerebroventricular brain region of normal mice to induce Aβ-associated cognitive impairments. During the auditory oddball paradigm, electroencephalograms (EEG) were recorded from frontal and parietal cortex of Aβ-infused and control mice. The event-related potentials (ERPs) elicited by auditory stimuli showed no significant difference in Aβ-infused mice compared to control mice. On the other hand, the differential ERP signature elicited by oddball sound stimuli was destructed in the Aβ-infused mice group. We noticed that ERP traces to standard and deviant tones were not significantly different in the Aβ group, while the control group showed differences in the amplitude of ERP components. In particular, the difference in the first negative component (N1) between standard and deviant tone, which indexes the sensory memory system, was significantly reduced in the parietal cortex of Aβ-infused mice. These findings demonstrate the direct influence of Aβ oligomers on the functional integrity of cortical areas *in vivo*. Furthermore, the N1 amplitude difference may provide a potential marker of sensory memory deficits in a mouse model of AD and yield additional targets for drug assessment in AD.

## Introduction

Alzheimer's disease (AD) is a neurodegenerative disease characterized by abnormal accumulation of amyloid-β (Aβ) and progressive impairments of cognitive abilities. When amyloid precursor proteins are cleaved by β- and γ-secretases on the membrane of neurons, Aβ peptides

funded by the Korean Government, No. 2017R1A2B3012659 (JHC), and the National Research Council of Science and Technology of Korea for the project, Development of Solution for Diagnosis, Treatment and Care System of Dementia, CRC-15-04-KIST (JHC); and the Korea Health Industry Development Institute (KHIDI), HI18C0836010018 (YK). This Work was performed at Center for Neuroscience, Korea Institute of Science and Technology (KIST). The funders had no role in study design, data collection and analysis, decision to publish, or preparation of the manuscript.

**Competing interests:** The authors have declared that no competing interests exist.

are released in the extracellular regions and begin to misfold into soluble oligomers and insoluble plaques [1]. Among several Aβ aggregate types, oligomers are considered to be responsible for major biochemical causes of neurodegeneration, such as neuroinflammation, synaptic/neuronal injury, neuronal ionic homeostasis breakdown, oxidative stress, and tau abnormalities in AD brains [2]. It is well known that increasing levels of neurotoxic Aβ oligomers in the brain impair cognitive behaviors and synaptic plasticity [3, 4]. As AD is a chronic disorder requiring timely accumulation of Aβ, transgenic animal models overexpressing human amyloid precursor proteins are often preferred in preclinical investigations of Aβ-associated pathophysiology [5]. However, these models are not suitable for Aβ oligomer-specific experiments due to the simultaneous and uncontrolled production of diverse pathogenic species, such as amyloid precursor proteins, Aβ (1–42) monomers, Aβ (1–40) monomers, Aβ oligomers, Aβ plaques, and enzymatic cleavage byproducts.

To conduct in vivo studies in an Aβ-species-controlled manner, Aβ-infused animal models are suggested [4]. Pathogen-induced rodent models are commonly used in neurodegenerative studies. Injection of scopolamine or 1-methyl-4-phenyl-1,2,3,6-tetrahydropyridine are well established methods to induce acute onsets of cognitive deficits and movement disorders, respectively, in rodents [6, 7]. Unlike these chemical pathogens that are injected intravenously, Aβ is injected directly into the intracerebroventricular (ICV) region of the brain with controlled aggregate species and peptide concentrations. Aβ injection rodent models acutely exhibit hippocampal-dependent learning and memory deficits in Y-maze, passive avoidance, fear conditioning, Morris water maze, and novel object recognition tests and Alzheimer-like pathological alterations such as decreased long-term potentiation, increased inflammation, decreased acetylcholine levels, activated astrocyte/microglia, and amyloid deposition [8–12]. In addition to the controllable Aβ conditions, benefits to bypass the ageing process of transgenic models for AD-like symptom and pathology onsets have allowed researchers to shorten the in vivo efficacy evaluation step of AD drug candidates directly regulating Aβ [9, 10, 12–17]. The Aβ-infused animal models are mostly investigated for less than a month since the peptide injection because the onset of AD-like phenotypes are promptly made, a recent study reported that memory deficits and synaptoxicity of mice became gradually worse in 40 days since the single injection of Aβ(1–42) [8].

Currently, multiple behavioral assays are available for testing hippocampal-dependent spatial memory (e.g., the Morris water maze), contextual memory (e.g., fear conditioning) or working memory (e.g., the Y-maze) functions [18] in AD animal models. These assays focus on memory loss, which is one of the most common signs of AD. However, the loss of memory is not the only AD symptom, and overreliance on these assays can contribute to the failure of translational studies across species. Other clinical symptoms related to brain functions, such as sensory processing, have not been investigated fully by the AD mouse research community due to the difficulty in assessing these functions in a mouse model. One potential research strategy is the use of test assays that are applicable in both humans and mice through the delivery of cross-species comparative parameters. In this regard, the use of electrophysiological responses might be a suitable means to assess cognitive deficits in a translational study of an AD mouse model.

Electrophysiological signals generated by neural circuitry seem to be preserved among species [19]. One of the most commonly used paradigms for measuring neural function is the presentation of sensory input, such as the auditory oddball paradigm. Event-related potentials (ERPs) have been investigated as biomarkers of cognitive decline and disease severity in patients with mild cognitive impairment (MCI) and Alzheimer's disease [20–24]. Auditory ERP traces have several positive and negative peaks after sound onset, representing a well-defined brain response to a sensory process. Generally, early ERP components, appearing as peaks

within ~ 200 ms, index rapid and automatic brain processes, while later components represent slower and more complex functions. The early components have been presumed to be suited for cross-species investigation since they represent automatic brain functions involving auditory discrimination without consciousness [25]. Previous studies suggested that rodents show auditory ERP components similar to humans with a systematically short latency [26, 27]. Umbricht et al. [26] presented the same auditory stimuli to both humans and mice and generated a formula for calculating the difference between latencies (Latency (human) = 1.67 * Latency (mouse) + 37.61 ms). The early sensory component of the auditory response includes an initial positive peak at approximately 50 ms and 20 ms (P1), an initial negative peak at approximately 100 ms and 40 ms (N1), and a second positive peak at approximately 200 ms and 120 ms (P2) in both humans and mice, respectively. The function of each ERP component has been investigated in a human study [28]. P1 is related to sensory gating, which is suppressed during repetitive auditory stimulation. N1 is linked to early attention and sensory memory-related variables and is related to detecting changes in sensory input [29–31]. N1 overlaps with the auditory mismatch negativity (MMN) specific to the oddball sound deviant from expected stimulus, which shows exaggerated and delayed negativity associated with the discrimination of deviant stimuli [32, 33]. The MMN classically is calculated as a negative deflection near the N1 period; it is a differential waveform obtained by subtracting the ERP for the repetitive tone from the ERP for the deviant tone. P2 is considered to reflect the endogenous neural mechanism linked to initial conscious awareness. Anomalies in these components have been reported in various brain disorders and their animal models [34–38]. Although consciousness-related late components of ERPs have been the main focus of investigations in Alzheimer's patients [22, 39, 40], several studies have shown a correlation between AD pathology and alterations in early components. Compared to that in normal older control subjects, the amplitude of the MMN has been shown to be attenuated in AD [20, 41], MCI [21, 24] and pre-symptomatic individuals who have mutations in an AD-related gene [42]. In mild AD, the N1 amplitude also been shown to decrease in response to both standard and deviant auditory stimuli [20]. Based on these previous results of MMN and N1 reductions, we hypothesized that novel sounds may be hard to perceive in Alzheimer's disease, resulting from impaired automatic attention to the deviant stimulus.

Clinical measurements of such ERP alterations have not yet been assessed in a mouse model of AD, whereas diminished ERP components have been observed in the mouse model for schizophrenia [38, 43, 44]. In the AD mouse model, EEG studies investigating brain dysfunction have been limited to analyses of spontaneous EEG signals. Moreover, spectral analysis of spontaneous EEG signals has yielded contradictory results in the AD mouse model, showing either an increase or decrease in the amplitude of delta or subdelta frequency signals in 5xFAD, PLB1 and APP/PS1 transgenic mice [45–47]. In this study, we aim to evaluate early components of the auditory ERP that represent novelty discrimination abilities in AD model mice. We characterized the auditory ERP response using an oddball paradigm in a mouse model mimicking AD pathology a result of Aβ injections into the ventricle to study its neurophysiological influence.

## Materials and methods

### Animals

For the generation of Aβ (1–42) infusion mice, 6-week-old Imprinting Control Region (ICR) male mice were purchased from Orient Bio Inc. (Seoul, Korea) and habituated for 4 days. Mice were maintained in a sterile laboratory animal breeding room at the Korea Institute of Science and Technology with stable temperature and humidity. Mice were exposed to a

controlled 12-hour light-dark cycle with food and water ad libitum. Nineteen mice were purchased for the study (n = 7 for vehicle group, n = 12 for Aβ group). After all experiments, animals were euthanized using carbon dioxide gas for over 10 minutes under deep anesthesia induced by intraperitorial injection of 2% avertin (250mg/kg). All animal experiments were performed in accordance with the National Institutes of Health guide for the care and use of laboratory animals (NIH Publications No. 8023, revised 1978). The animal studies were approved by the Institutional Animal Care and Use Committee of Korea Institute of Science and Technology (Approval Number: 2016–035).

## Aβ-infused mice model

We produced the model according to previously demonstrated method [4]. For each injection, 100 μM Aβ (1–42) was prepared (10% DMSO [dimethyl sulfoxide] in PBS). All 100 μM Aβ samples were incubated at 37˚C for 7 days. After incubation, mice were anaesthetized with 4% avertin(250mg/kg) and assessed the foot or tail pinch response to confirm adequate anesthesia. While anesthetized, mice were placed on a warm mat to maintain its body temperature. 5 μL of vehicle (10% DMSO in PBS), or Aβ solution was injected into the lateral ventricle by intra-cerebroventricular (ICV) injection [4]. Briefly, using their thumb and index finger, we tightly hold down the mouse skin of the forehead and drag the skin behind to minimize skull movement under skin. Without removing the skin, we positioned and inserted the needle into the lateral ventricle (3.8 mm depth). Then we slowly inject 5 μL of vehicle or Aβ solution over 5 s and wait 3 to 5 s before removing the syringe for diffusion. After injection, mice were placed in cage above the warm pad and watched for abnormalities until it regains consciousness. We postoperatively monitor the mouse behavior for several days.

## Y-maze test

The Y-maze apparatus was constructed with black plastic and composed of three equally spaced arms (40 L × 10 W × 12 H cm). The mice were placed at the end of one arm and allowed to freely explore their surroundings for a 12-min session. An arm entry was defined as the tip of the tail of the mouse entering the arm entirely. Entries of all mice were recorded manually. A different entry than the previous two entries was defined as an alternation, and the following equation was used to calculate spontaneous alternation behavior.

$$\%\text{alternation} = 100 \times [(\text{number of alternations})/(\text{total number of arm entries} - 2)]$$

## Surgery for EEG electrodes

To implant the microscrew electrodes, surgical procedures were performed under deep anesthesia with intraperitoneal injection of a ketamine (120 mg/kg) and xylazine (6 mg/kg) cocktail. Five minutes after i.p. injection, mouse toe and tail pinch was done to check whether the anesthesia is sufficient. When mouse did not show any reaction to the toe and tail pinch, the animal placed on to the stereotaxic apparatus (Kopf Instruments, Tujunga, CA, USA). Sterilized microscrew electrodes (0.8 mm diameter, Asia Bolt, South Korea) were fixed onto the skull surface of the frontal (anteroposterior, 2 mm; mediolateral, 1 mm) and parietal cortex (anteroposterior, 2 mm; mediolateral, 2 mm), with ground/reference electrodes on the occipital bone above the cerebellum. The electrode coordinates were determined according to the mouse atlas [48]. Dental cement (VertexTM Self-Curing, Vertex-Dental, Netherlands) was applied to secure the position of the electrodes. After surgery, the incision was closed with sterile suture and antibiotic ointment (sodium fusidate, 20mg/g) was applied to prevent infection.

Mice were treated with analgesic ointment (ketoprofen, 30mg/g) and underwent one week of recovery.

## Auditory oddball test

The test was performed in a transparent cylinder (10 cm in diameter, 25 cm in height), which was placed in the center of Faraday cage (55 cm x 60 cm x 65 cm, light and sound proof). A 60 dB white noise was presented throughout the experiment as background noise (White noise generator, San Diego Instrument, CA, USA). For acclimation, we placed the bedding from the home cage at the cylinder floor and left the animal for at least 15 min prior to the experiment. The auditory oddball test was composed of two pure tone sound stimuli (2 kHz as the standard tone and 4 kHz as the deviant tone, 80–85 dB, 10 ms) presented via four surround speakers (Dongguan edifier technology, China). The temporal sequence and frequency were predetermined by voltage output based on a custom-built MATLAB program (Mathworks, Inc. Natick, MA, USA). The digital output was sent to the speaker and high-level input port in the amplifier via a digital-to-analog converter (NI 9263 measurement system, National Instruments, TX, USA). The ratio of standard to deviant tones was 9 to 1, where the deviant tone was presented randomly with the restriction of having >3 standard tones between two deviant tones. The whole session consisted of 1000 stimulus presentations with variable interstimulus intervals (1–1.5 s) and lasted approximately half an hour (S1 Fig).

## EEG acquisition

EEG data were collected in a Neuroscan SynAmps2 amplifier system (Compumedics, Charlotte, NC, USA) at a 2 kHz sampling rate. The impedance of each electrode was kept below 300 kΩ, and online filtering (60 Hz notch and 1–200 Hz bandpass) was applied.

## ERP analysis

Data was analyzed using the MATLAB (Mathworks Inc., Natick, MA, USA). We obtained ERP by following procedure: First, we filtered the EEG with a band-pass filter (cut off frequencies = 0.5 and 59 Hz, FIR filter type). Next, we extracted 1 s of epochs from -0.4 s to 0.6 s with respect to the auditory stimulus. Any epoch with high voltage with absolute voltage over 1 mV was considered to be contaminated and then excluded in further analysis. Then, we corrected the baseline by subtracting the mean EEG values in the prestimulus period from -100 to 0 ms. Lastly, the ERP was obtained by average individual epochs of EEG separately for the standard and deviant stimuli. To balance the numbers of two conditions, we used only the epoch with the standard stimulus preceding the epoch with deviant stimulus, of which number is roughly 100 epochs per mouse.

The peaks of ERP were obtained based on peak-detection algorithm. Previous studies reported that the auditory ERP in mice typically show peaks at latencies within 120 ms after stimulus onset. Each peak was considered equivalent to human P1, N1, and P2 components. We detected the ERP peaks for individual mice using peak detection algorithms based on max-min amplitude in the window of interest. Specifically, P1, N1 and P2 peaks were defined at maximum points between 10 and 25 ms, minimum locations between 25 and 45 ms, and maximum locations between 45 and 200 ms, respectively. The ranges were modified from the method used in previous mice studies [27]. All detected peaks were visually confirmed. The peak detection results are summarized in Table 1.

**Table 1. Summary of amplitude and latency of ERP component peaks to deviant and standard sound.**

| ERP component | Region | Amplitude (μV) | | | | Latency (ms) | | | |
| --- | --- | --- | --- | --- | --- | --- | --- | --- | --- |
| | | Control | | Aβ-infusion | | Control | | Aβ-infusion | |
| | | Standard | Deviant | Standard | Deviant | Standard | Deviant | Standard | Deviant |
| P1 | Frontal | 3.1±5.1 | **33.4±11.7**\* | 15.6±6.2 | 30.3±11.0 | 14.6±1.9 | 16.7±0.4 | 15.7±1.7 | 17.1±1.6 |
| N1 | Frontal | -25.0±5.5 | **-60.7±5.2**\*\* | -27.5±9.1 | -50.4±5.4 | -35.1±1.4 | -35.6±1.3 | -35.8±1.6 | -35.0±1.1 |
| P2 | Frontal | 15.8±4.7 | **37.8±7.9**\* | 22.8±6.7 | 35.6±6.5 | 111.9±9.3 | 101.2±14.6 | 87.9±9.7 | 83.4±15.9 |
| P1 | Parietal | 9.4±6.5 | 51.0±17.9 | 29.9±13.1 | 53.9±17.2 | 17.0±2.0 | 18.0±0.4 | 16.8±1.9 | 17.2±1.6 |
| N1 | Parietal | -21.6±6.7 | **-66.0±8.4**\*\* | -16.3±7.4 | -37.7±15.4 | -34.4±1.3 | -33.8±1.3 | -35.7±3.3 | -34.8±2.3 |
| P2 | Parietal | 14.1±5.4 | **40.8±8.3**\* | 29.4±10.6 | 35.1±10.5 | 104.2±9.0 | 103.2±16.7 | 91.4±17.0 | 92.2±17.4 |

Data are represented as the mean ± s.e.m. Component values showing significant differences between responses to standard and deviant tones are marked with bold and asterisks (Student's t-test, \*p < 0.05; \*\*p < 0.01)

### Statistical analysis

A Lilliefors normality test was used to test the null hypothesis that the data exhibited a normal distribution. ERP waveforms and detected ERP components were compared between groups or conditions by a Student's t-test or Kruskal-Wallis test where appropriate. A two-way ANOVA was performed to test effect of group (vehicle vs. Aβ-infusion) and stimulation conditions (standard vs. deviant) and their interaction. A sign test was used to test whether the negative deflection near N1 of the differential ERP waveforms showed significant negativity compared with 0. An alpha level of 0.05 was considered a significant result.

## Results

### Behavioral performance

Prior to electrophysiological recordings, vehicle (6-week-old males, N = 7) or Aβ (6-week-old males, N = 12) injected mice were subjected to Y-maze spontaneous alternation tests to assess working memory ability on the third days of injection (Fig 1A). Total arm entries and percentage of alternations were calculated to assess locomotion and spatial working memory abilities, respectively. We found that the alternation rate was significantly reduced in the Aβ group compared to the vehicle group (Student's t-test, p = 0.007, Fig 1B), while locomotion was not affected (p = 0.45, Fig 1C). The working memory of 6-week-old ICR male mice was confirmed to be reduced by the ICV injection of Aβ.

### ERP waveforms

At one month after ICV injection (32.1 ± 2.1 days, ranging from 29 to 36 day after injection), EEGs were recorded from frontal and parietal cortex during presentation of auditory oddball paradigm. Due to implantation of EEG electrodes and recovery procedures, there was a three-week gap between behavioral and EEG experiments. Fig 2 presents the event-related potentials (ERP) elicited by standard (2 kHz, 90%) and deviant (4 kHz, 10%) sound stimuli averaged over vehicle (N = 6) and Aβ-infused (N = 7) mice. The time windows showing significant differences between responses to standard and deviant sound presentations are marked by a gray shadow overlapping the ERP plots (Student's t-test, p < 0.05). In the vehicle group, the deviant ERP was significantly different from the standard ERP. In the frontal cortex, there was a significant difference between the two types of ERP in the very early (10–22 ms), early (27–39 ms), intermediate (73–102 ms), and late periods (150–300 ms), which correspond to time windows for the P1, N1, P2, and P3 peaks in rodents [27]. In the parietal cortex, the two ERPs were

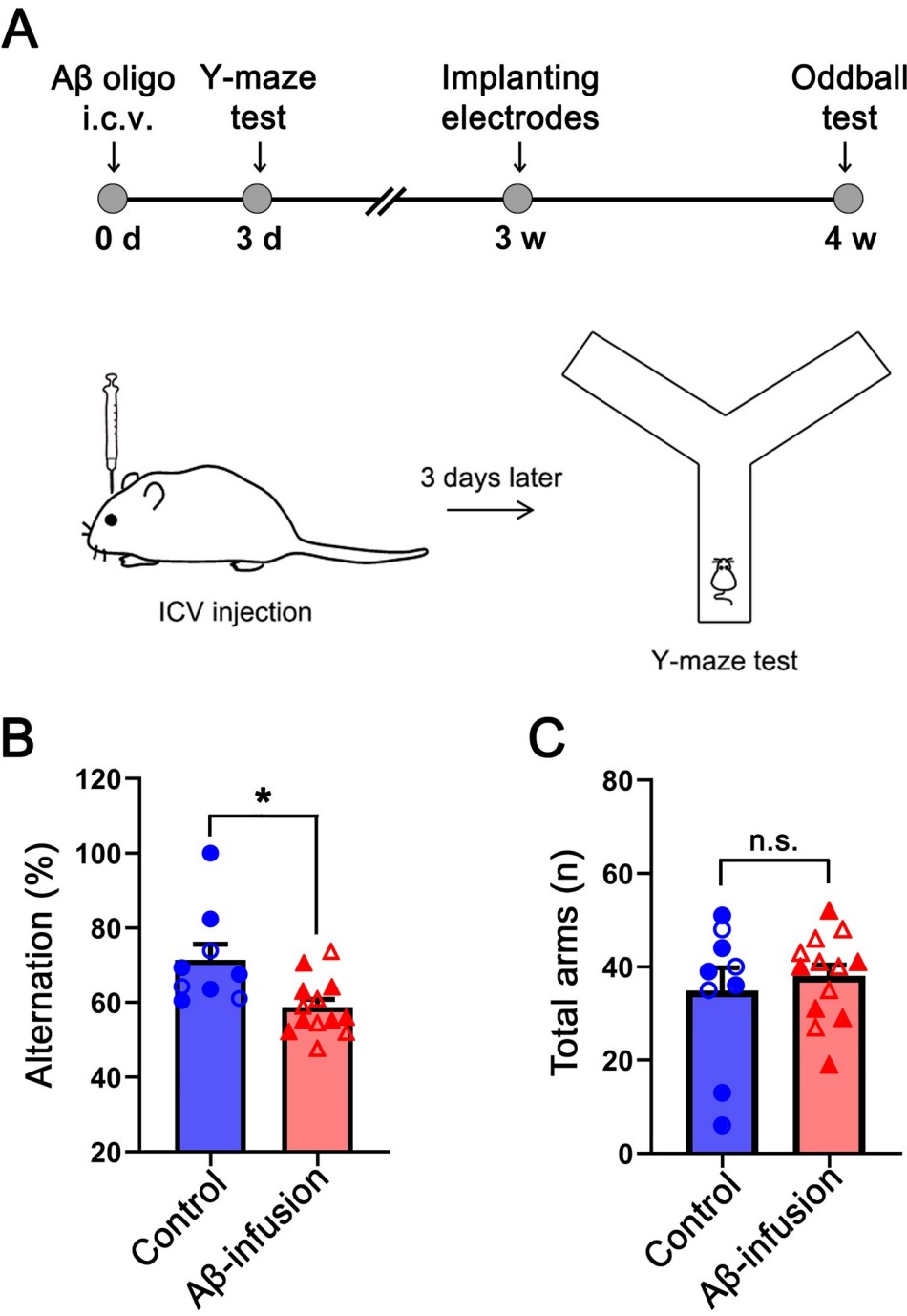

**Fig 1. Y-maze behavioral test.** (A) The experimental time-line for all experimental procedures for Y-maze spontaneous alteration test and auditory oddball ERP test (up). Detailed study design for assessing working memory ability following intracerebroventricular (ICV) injection of Aβ using Y-maze(bottom). (B) Mean percent alteration of Aβ-infused (red) and control (blue) groups is depicted in the bar graph with error bars representing SEM. The value of each individual is indicated by a dot on the bar graph. Filled dots represent individual mice used both behavioral test and auditory oddball test. (C) The bar graph represents the mean and SEM of total arm entries in the Aβ-infused and control groups (Student's t-test $p < 0.05$ as significance level). d: day; w: week.

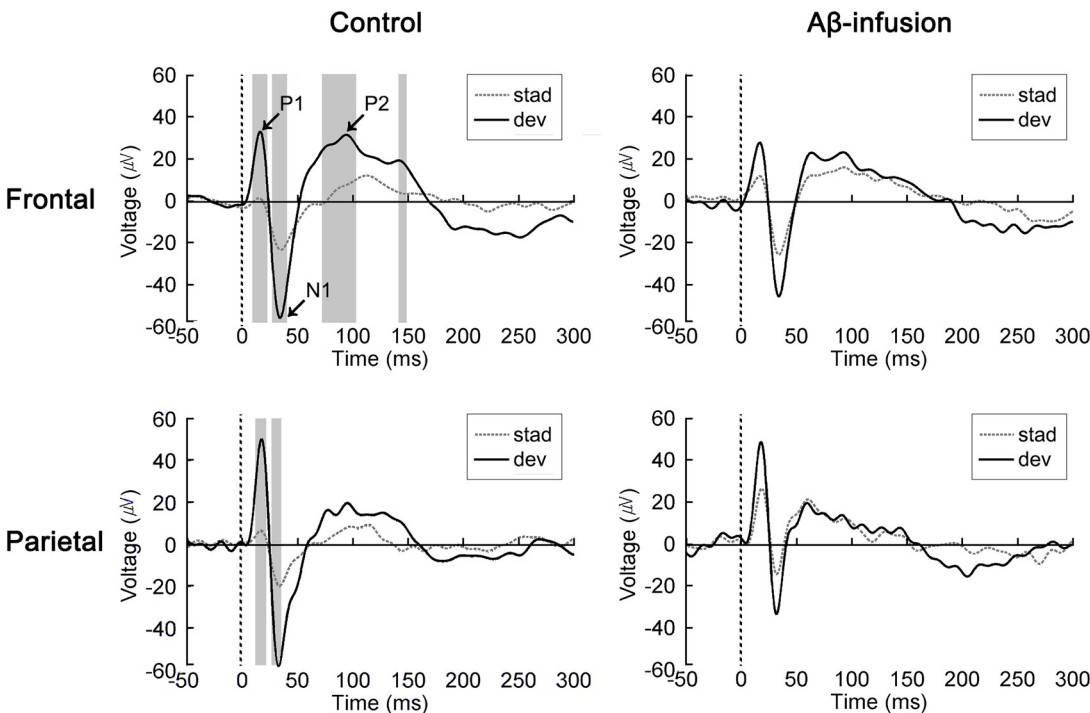

**Fig 2. ERP responses to standard and deviant tones.** Grand-averaged ERP waveforms elicited by standard (dotted line) and deviant (solid line) tones during auditory oddball paradigm in frontal (top) and parietal cortex (bottom) in control (left) and amyloid-beta infused (Aβ-infusion, right) mice. Significantly different period between standard and deviant ERP highlighted by gray shade (Student's t-test, $p < 0.05$). Tones were presented at time 0 ms on the x-axis. Peaks of early ERP components (P1, N1, and P2) are marked by arrows in the top-left panel. stad, standard tone; dev, deviant tone.

significantly different during the very early (12–22 ms) and early periods (27–35 ms). Although the averaged ERP traces for the Aβ-infusion group were enhanced in response to the standard tone and reduced in response to the deviant tone compared to the vehicle group, the difference was not statistically significant between the groups. (S2 Fig, Student's t-test, p > 0.05).

The MMN classically is calculated as a negative deflection near the N1 period of differential waveform obtained by subtracting the ERP for the repetitive tone from the ERP for the deviant tone. We compare the differential ERP waveforms for groups. Fig 3 presents the differential ERP traces obtained by subtracting the standard ERP from the deviant ERP in frontal and parietal cortex averaged over vehicle (N = 6) and Aβ (N = 7) infused mice. The grand-averaged differential ERP of the Aβ-infusion group was reduced in both frontal and parietal cortex. However, the differential ERP waveforms were not significantly different between groups (Fig 3, Student's t-test, p > 0.05).

### Peak amplitude of early components and MMN-like activity

The amplitude and latency of the P1, N1, and P2 peaks were determined by maximum or minimum peak detection and are summarized in Table 1. Consistent with the ERP wave forms, a significantly different amplitude in the early components between responses to the deviant and standard tones was observed in the control group, while this difference was destructed in both the frontal and parietal ERP of the Aβ-infusion group (Table 1). In order to analyze the effects of stimulus type and groups, we performed a two-way ANOVA on the peak values of

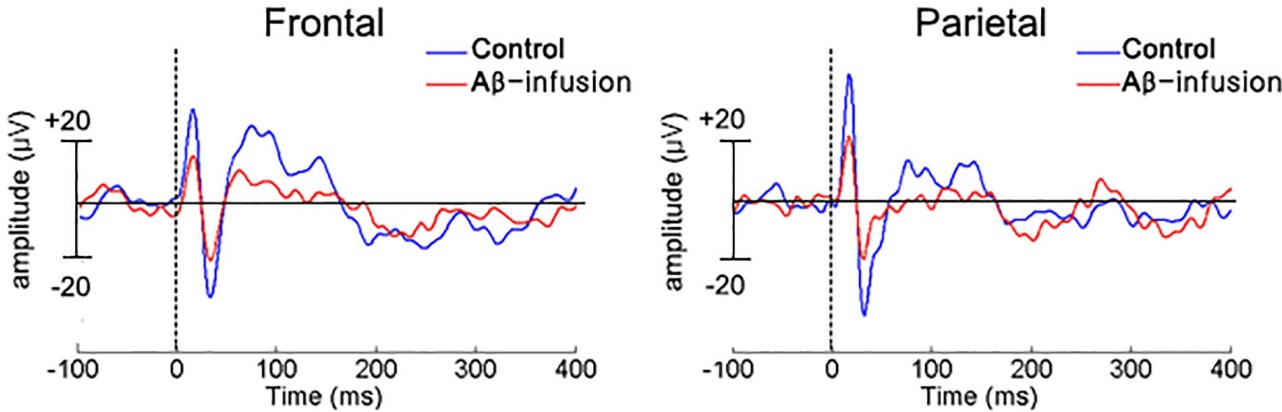

**Fig 3. Differential ERP waveforms between standard and deviant tones.** Grand-averaged differential ERPs between standard and deviant tone measured in frontal cortex (A) and parietal cortex (B) in control (blue-line) and amyloid-beta infused (Aβ-infusion, red-line) mice. Significantly different periods were not observed when comparing control and Aβ-infusion mice.

P1, N1, and P2, and groups from the frontal ERP. We found a main effect in the stimulus condition (deviant vs standard, $F_{(1,22)}$ = 6.25, 19.07, and 7.03 for P1, N1, and P2, respectively, all $p < 0.05$). On the other way, neither the group effect nor the stimulation type x group interaction was found to be significant. The differently elicited ERP between standard and deviant sound were known to relate with auditory sensory memory which important in ability to automatically attend to novel environment. To report the marginal tendency of the reduced differential ERP in Aβ-infused mice, we calculated the amplitude changes to deviant tones in early components, subtracting the amplitude of early components detected in the standard ERP from that of the deviant ERP. Since the shape and latency of the ERP responses to standard and deviant tones are identical in mouse, the MMN-like component in mice is supposed to be the same as the difference in N1 amplitude between deviant and standard ERPs. The waveform of the vehicle group produced robust negative deflection via sign tests (difference from zero) performed on the peaks near the N1 period (20–80 ms) in both frontal and parietal cortex (sign test, $p = 0.03$ same in frontal and parietal cortex), while the Aβ-infusion group failed to show statistically significant negative peaks in both cortices (sign test, $p = 0.16$ for frontal cortex, $p = 0.22$ for parietal cortex). In frontal cortex, the maximum negativity was -36.5 ± 4.4 and -25.2 ± 9.6 μV in the vehicle and Aβ-infusion groups, respectively. In parietal cortex, the negative peak was -45.2 ± 7.7 and -25.0 ± 9.1 μV in the vehicle and Aβ-infusion groups, respectively. The differences in early components between the responses to standard and deviant tones were smaller in Aβ-infused mice than in controls in both frontal and parietal cortex. However, a significantly reduced amplitude was found only in the differential N1 component in the parietal cortex (Fig 4, $p = 0.03$, Kruskal-Wallis test).

## Discussion

Recent studies shed light on neuronal dysfunction induced by amyloid-beta levels [49]. In accordance with this perspective, we inspected whether the increase of amyloid β disrupts neural activity related to sensory processing in a mouse model of Aβ infusion. Our aim was to determine whether the auditory oddball paradigm suggested in human patients to diagnose AD, could indicate the severity of AD pathology in the mouse. In our AD model, Aβ-infused mice, the auditory ERP did not discriminate deviant tones from the standard tones. Compared to the control group, Aβ-infused mice tended to exhibit a reduced N1 component in response

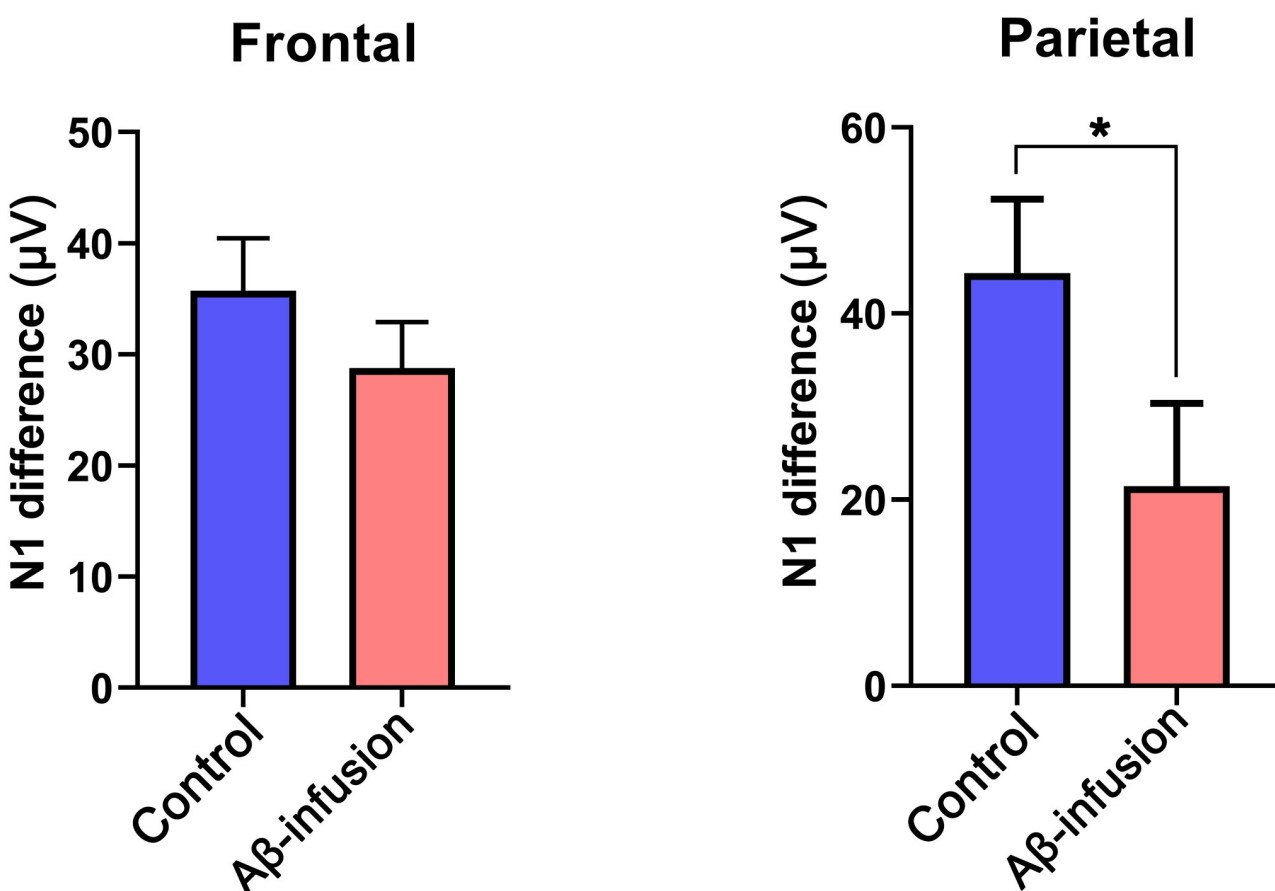

**Fig 4. N1 difference between standard and deviant tones.** Bar graphs represent the mean amplitude difference in N1 between standard and deviant tones in frontal (left) and parietal cortex (right) in control (blue bar) and Aβ-infusion mice (red bar). The error bars represent the SEM. Statistical analysis was performed with the Kruskal-Wallis test ($^*$ $p < 0.05$).

to deviant sound stimuli in both frontal and parietal regions. Only the parietal region showed a significantly attenuated N1 differential response between deviant and standard tones. The amplitude of the N1 component was correlated with impaired Y-maze test performance.

The shape and latency of auditory ERPs in this study are consistent with previous oddball studies conducted in rodents [44, 50]. The shape of ERP components in the rodent is similar to that in humans, with a systematically short latency [26, 27]. Additionally, researchers have debated whether the ERP component specific to novel sound is comparable across species. The mismatch negativity (MMN), which indexes novelty in the auditory environment, is represented by a negative peak around the N1 latency in the differential waveform generated by subtracting the standard tone ERP from the deviant tone ERP. In human, MMN component is distinct from N1 although N1 contribute to shape and amplitude of difference waveform between ERP to standard and deviant tone [51, 52]. Without the adaption process of N1 by repetitive stimuli, MMN is presented when expected stimulus does not matched to the presented one [52, 53]. During oddball test, participants was thought to create a model of the regularities in the acoustic environment by repetitive sound, and expect continuation of same stimulus. MMN induced by oddball sound is regarded as a signal for the regularity violation or the prediction error [54].

Although the amplitude of N1 is larger in response to the deviant tone, the shape and latency of the ERP responses to standard and deviant tones are identical in mouse studies [55]. Therefore, the MMN in mice is supposed to be the same as the difference in N1 amplitude between deviant and standard ERPs. In this study, we calculated the N1 difference and considered it to be analogous but not exactly the same as the human MMN since the N1 difference has been known to partially account for the MMN wave in humans [56]. We found significantly reduced N1 differences in the parietal cortex in the AD mouse model. Additionally, although not statistically significant, the standard-deviant differences at all components in both frontal and parietal regions tended to be attenuated in the AD mouse model, which can be inferred from the lack of significant differences in the ERP responses to standard and deviant tones in terms of the whole trace in both frontal and parietal regions.

Similar to the reduced N1 difference in our mouse model of AD, MCI and Alzheimer's patients generally exhibit significantly lower MMN amplitudes than do healthy controls. Previous MMN studies with contradictory results reveal that the interstimulus interval could be important in whether a significantly reduced MMN is observed between AD and control groups. When a short interstimulus interval (~ 1 s) was adopted, the MMN amplitude was intact compared to the control group [41, 57]. A longer interstimulus interval systematically decreases N1 adaptation to repetitive sound and therefore reduces MMN amplitude. The MMN has been shown to disappear when the interval was 8–10 s long [35]. Pekkonen et al. used various interstimulus intervals and reported that the MMN amplitude was normal in AD patients when the interstimulus interval between deviant and repetitive tone was less than 1 s; however, it more sharply decreased as a function of the interstimulus interval in AD. An intact MMN at short intervals was interpreted as the normal formation of the auditory sensory memory trace in AD patients, but the retention period of the sensory memory was as short as 1 s to detect an environmental change in sound stimulation. In this study, we used a maximum of 1.5 seconds for the interstimulus interval (1–1.5 s), which is longer than previous oddball studies in rodents that adopted a 500 ms interval [27, 38]. Due to the lack of previous investigations of the oddball response in rodents, we cannot determine whether the interstimulus of 1.5 s in mice is optimal for dissociating AD and the control group. Our results show a trend in which the differential response to deviant and standard tones were attenuated in both frontal and parietal cortex in Aβ-infused mice. However, only the N1 difference in parietal cortex showed a significant decrease. The marginally significant results suggest that the 1.5 s interval is not long enough to observe sensory memory deficits in the AD mouse model. The deviant-specific response itself vanishes if the interval is too long to retain sensory memory in the brain, whereas the deviant-specific response is not altered in AD if the interstimulus interval is too short. Therefore, further research is necessary to determine the optimal interval to differentiate AD model mice from control group.

An impaired oddball response could be induced by Aβ-related synaptic dysfunction. N-methyl-D-aspartate receptors (NMDAR) and the induction of long-term potentiation (LTP) are required for auditory discrimination [44, 58]. Reduction of NMDA receptors on the neural surface [59] and blocking of LTP both in vitro and in vivo [60] have been reported as synaptic impairments induced by Aβ. Our AD mouse model was produced by direct injection of soluble oligomeric Aβ into the cerebrospinal fluid (CSF) of the ventricle. Because soluble oligomeric Aβ is able to easily transfer via brain parenchyma and neuronal connections [61], Aβ in CSF may affect global brain tissue due to CSF circulation. We speculate that neural circuits generating an oddball response would be globally affected by the soluble Aβ, resulting in NMDAR receptor abnormalities. Previous studies showed that the integrity of the NMDAR system could be well represented by MMN component in both human and rodents. MMN amplitude was attenuated by NMDA antagonist, ketamine, and correlated with severance of

psychotic responses after ketamine injection [62]. Studies in schizophrenic patients extensively investigated the link between NMDAR hypofunction and MMN attenuation since they are prominent features in schizophrenia [63]. Conversely, modulator of NMDAR function increases MMN in patient [64]. The MMN-like component was also altered by NMDR blocker such as MK-801 or ketamine during the oddball paradigm in rat and mice [65]. Similar to human, the NMDR blockers generally reduced MMN-like component in rodents [44, 66]. Moreover, direct neurotoxicity of Aβ has been confirmed in brain regions relevant to sensory processing and memory consolidation. Calcium-dependent neurotoxic events and oxidative injury following Aβ administration may result in acute cognitive impairments [67].

In addition to behavioral characterization of AD, mouse ERP profiles of sensory processing may serve as valuable cognitive measures comparable to human cognition, facilitating diagnosis and assessment of drug effects and disease progression in the AD mouse model.

## Supporting information

**S1 Fig. Auditory oddball paradigm.** Sound stimuli for the auditory oddball test were described in diagrammatic depiction. Deviant (black) and standard (gray) tones were randomly presented for 10 ms with 1:9 ratio through the speakers around mice. Interstimulus intervals randomly changed in the range from 1 to 1.5 s.
(TIF)

**S2 Fig. ERP waveforms in response to standard and deviant tones.** Grand-averaged ERP traces for control (blue-line) and Aβ-infusion (red-line) group were compared. ERP waveforms elicited by standard (top) and deviant (bottom) tones in frontal (left) and parietal (right) regions were presented. Sounds were presented at time zero. In the top-left panel, arrowheads pointed the P1, N1 and P2 components. A significant difference in ERP time trace between control and Aβ-infusion groups were not detected (Student's t-test, $p < 0.05$).
(TIF)

## Author Contributions

**Conceptualization:** YoungSoo Kim, Jee Hyun Choi.

**Data curation:** Bowon Kim, Jisu Shin.

**Funding acquisition:** YoungSoo Kim, Jee Hyun Choi.

**Investigation:** Bowon Kim, Jisu Shin.

**Methodology:** Bowon Kim, Jisu Shin.

**Supervision:** YoungSoo Kim, Jee Hyun Choi.

**Visualization:** Bowon Kim.

**Writing – original draft:** Bowon Kim, Jisu Shin, YoungSoo Kim, Jee Hyun Choi.

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
