## [Decision Letter · Decision Letter 0]

5 Jan 2020

PONE-D-19-32332

Destruction of ERP responses to deviance in an auditory oddball paradigm in amyloid infusion mice with memory deficits

PLOS ONE

Dear Dr. Choi,

Thank you for submitting your manuscript to PLOS ONE. After careful consideration, we feel that it has merit but does not fully meet PLOS ONE’s publication criteria as it currently stands. Therefore, we invite you to submit a revised version of the manuscript that addresses the points raised during the review process.

We would appreciate receiving your revised manuscript by Feb 19 2020 11:59PM. To enhance the reproducibility of your results, we recommend that if applicable you deposit your laboratory protocols in protocols.io, where a protocol can be assigned its own identifier (DOI) such that it can be cited independently in the future. For instructions see: http://journals.plos.org/plosone/s/submission-guidelines#loc-laboratory-protocols

We look forward to receiving your revised manuscript.

Kind regards,

Dong-Gyu Jo, Ph.D

Academic Editor

PLOS ONE

Journal Requirements:

2. To comply with PLOS ONE submissions requirements, in your Methods section, please provide additional information on the animal research and ensure you have included details on (1) methods of sacrifice, (2) methods of anesthesia and/or analgesia, and (3) efforts to alleviate suffering. Please ensure that you have provided this information for all parts of your study, including the icv injections.

I have read the journal's policy and the authors of this manuscript have the following competing interests

a) Please provide an amended Funding Statement that declares *all* the funding or sources of support received during this specific study (whether external or internal to your organization) as detailed online in our guide for authors at http://journals.plos.org/plosone/s/submit-now.  

b) Please state what role the funders took in the study.  If any authors received a salary from any of your funders, please state which authors and which funder. If the funders had no role, please state: "The funders had no role in study design, data collection and analysis, decision to publish, or preparation of the manuscript."

Reviewers' comments:

Reviewer's Responses to Questions

**Comments to the Author**

1. Is the manuscript technically sound, and do the data support the conclusions?

Reviewer #1: Yes

Reviewer #2: Yes

2. Has the statistical analysis been performed appropriately and rigorously? 

Reviewer #1: Yes

Reviewer #2: Yes

3. Have the authors made all data underlying the findings in their manuscript fully available?

Reviewer #1: Yes

Reviewer #2: Yes

4. Is the manuscript presented in an intelligible fashion and written in standard English?

Reviewer #1: Yes

Reviewer #2: Yes

5. Review Comments to the Author

Reviewer #1: The authors conducted an auditory oddball test on the Alzheimer’s disease mouse model with infusing amyloid-beta intra-ventricle. They present data showing amyloid beta eliminated the difference of ERP signature, and showing the difference of MMN in A-beta group for the first time. They suggest this study may provide a markers of sensory memory in mouse AD model.

This manuscript describes a technically sound data and the experiments conducted rigorously. The conclusion is well drawn from data.

This manuscript can be re-evaluate to publish after some minor revisions performed.

Minor revision

1. affiliation has typo.

2. line7, ‘Alzheimer’s mouse model’ may change to “Alzheimer’s disease mouse model”

3. line 138, “method3” may be typo.

4. line 204, “studies27” may be type.

5. line 210, character “beta” is missing

6. recommend to be edited by native speaker.

Reviewer #2: The manuscript reports an interesting study of an Aβ infusion mouse model of Alzheimers Disease (AD). They investigate behavioural measures of spatial memory using the Y-maze task and electrophysiological measures of sensory memory using the oddball paradigm comparing deviant and standard responses. They report impaired performance of the AD mouse model on the Y- maze task as well as evidence that deviant responses in the oddball paradigm and perhaps mismatch like response are also reduced in the AD mice compared with the control mice.

1. The manuscript is clearly written, however this could be developed further in particular to tie together the interaction of the abeta oligomers with the likely interaction point at the level of the neuron that is then assayed by the electrophysiological tests – and how this correlates with assaying.

2. Was any work done to corroborate the distribution of the Abeta oligomers are ICV (whether within this study of previously)?

3. How do the concentrations and timings and electrophysiological findings correlate or differ from work done using abeta oligomer incubations in brain slice (rodent), or biopsy derived human material? This is important in the overall experimental rationale.

6. PLOS authors have the option to publish the peer review history of their article (what does this mean?). If published, this will include your full peer review and any attached files.

Reviewer #1: No

Reviewer #2: No

---

## [Author Response · Author response to Decision Letter 0]

9 Feb 2020

Reviewer #1

1. affiliation has typo.

We deleted the typo.

2Division of Bio-Medical Science & Technology, KIST School, Korea University of Science and Technology, Seoul, Republic of Koreaparentheses

2Division of Bio-Medical Science & Technology, KIST School, Korea University of Science and Technology, Seoul, Republic of Korea

2. line7, ‘Alzheimer’s mouse model’ may change to “Alzheimer’s disease mouse model”

We changed the sentence as you suggested. 

an Aβ-infused Alzheimer’s mouse model

an Aβ-infused Alzheimer’s disease mouse model

3. line 138, “method3” may be typo.

It was a style error for referencing. We corrected the error.

method3 was changed to

method [4].

4. line 204, “studies27” may be type.

studies27 was changed to 

studies [27]

5. line 210, character “beta” is missingWe fixed the error. 

 A�- was changed to Aβ-

Reviewer #2:

1. The manuscript is clearly written, however this could be developed further in particular to tie together the interaction of the abeta oligomers with the likely interaction point at the level of the neuron that is then assayed by the electrophysiological tests – and how this correlates with assaying.

[Reply] We agree with the reviewer’s opinion of adding value by collimating the pathological and electrophysiological changes by A infusion. Since the pathological data can be obtained after the experiment, it would be desirable to record EEG as close as possible to the injection time point. But the 4-week delay in EEG recording was inevitable to reach sufficient mouse welfare and health conditions required for obtaining reliable EEG responses. For instance, differential ERP is known to be reduced in inattentive or fatigued or unhealthy animals. Hence, we endowed animals 3 weeks before implantation surgery and another week before recording, for full recovery from infusion and implantation surgery, respectively. We believe that the Aβ oligomer injection influences the central nervous system in a devastating way. It is known that Aβ infusion induces an irreversible change in brain tissue: Cardinal features of Aβ infusion include synapse loss, tau hyperphosphorylation, astrocyte and microglial activation, and most importantly, neurofibrillary tangle formation, which is irreversible (Forny-Germano et al., J Neurosci, 2014). 

Our study shows an irreversible influence of ICV injection fingerprinted in the ERP results, which has a translational value because of its availability for cross-species comparison between human patients and the mouse model. 

2. Was any work done to corroborate the distribution of the Abeta oligomers are ICV (whether within this study of previously)?[Reply] We have reported a brain section-level evidence of A infusion in brain tissue in our previous methodology paper (Kim et al., J. Vis. Exp., 2016). However, the propagation of infusion effect over time through various brain regions has not been studied yet. We expect that the devastating effect of A infusion will not be restricted to the injection site since it affected differential ERP, engaging both bottom-up and top-down cortical circuits. 

3. How do the concentrations and timings and electrophysiological findings correlate or differ from work done using abeta oligomer incubations in brain slice (rodent), or biopsy derived human material? This is important in the overall experimental rationale.

[Reply] We appreciate this comment. As far as we know, a direct comparison between biopsy-derived human material and brain slice of infusion model has not yet been reported. Notwithstanding, the Aβ-ICV injection model has been widely investigated with variations of Aβ aggregate species, Aβ isomers, injection numbers, and base animal models accompanied by behavior tests. However, it is hard to find complete datasets of pathology and behaviors in terms of timing and concentration. Nonetheless, several research articles (cited in our manuscript) used similar experimental conditions to ours (~ 2 ug concentration, one-month delay after injection in observation). Here are the corresponding papers.

Scientific Reports (2015) 5, 11708 reports that memory deficits and synaptoxicity of mice became gradually worse over 3 and 40 days since the single injection of Aβ(1-42) 3 ug.

Journal of neuroscience research (2010) 88, 2923-2932 reports long-term potentiation changes with fear conditioning and CaN activity results by CV injection of Aβ oligomers in lower amounts than those of our experimental design.

Neuroscience letter (1994) 170, 63-66 reports behavioral alterations in water-maze by varied concentrations of Aβ monomers in lower amounts than those of our experimental design.

---

## [Decision Letter · Decision Letter 1]

26 Feb 2020

Destruction of ERP responses to deviance in an auditory oddball paradigm in amyloid infusion mice with memory deficits

PONE-D-19-32332R1

Dear Dr. Choi,

We are pleased to inform you that your manuscript has been judged scientifically suitable for publication and will be formally accepted for publication once it complies with all outstanding technical requirements.

With kind regards,

Dong-Gyu Jo, Ph.D

Academic Editor

PLOS ONE

Additional Editor Comments (optional):

Reviewers' comments:

Reviewer's Responses to Questions

**Comments to the Author**

1. If the authors have adequately addressed your comments raised in a previous round of review and you feel that this manuscript is now acceptable for publication, you may indicate that here to bypass the “Comments to the Author” section, enter your conflict of interest statement in the “Confidential to Editor” section, and submit your "Accept" recommendation.

Reviewer #1: All comments have been addressed

Reviewer #2: All comments have been addressed

2. Is the manuscript technically sound, and do the data support the conclusions?

Reviewer #1: Yes

Reviewer #2: Yes

3. Has the statistical analysis been performed appropriately and rigorously? 

Reviewer #1: Yes

Reviewer #2: Yes

4. Have the authors made all data underlying the findings in their manuscript fully available?

Reviewer #1: Yes

Reviewer #2: Yes

5. Is the manuscript presented in an intelligible fashion and written in standard English?

Reviewer #1: Yes

Reviewer #2: Yes

6. Review Comments to the Author

Reviewer #1: (No Response)

Reviewer #2: Authors have addressed all concerns and provide additional information that support the conclusion of their study.

7. PLOS authors have the option to publish the peer review history of their article (what does this mean?). If published, this will include your full peer review and any attached files.

Reviewer #1: No

Reviewer #2: No

---

## [Editor Report · Acceptance letter]

2 Mar 2020

PONE-D-19-32332R1 

Destruction of ERP responses to deviance in an auditory oddball paradigm in amyloid infusion mice with memory deficits 

Dear Dr. Choi:

I am pleased to inform you that your manuscript has been deemed suitable for publication in PLOS ONE. Congratulations! Your manuscript is now with our production department. 

With kind regards,

on behalf of

Dr. Dong-Gyu Jo 

Academic Editor

PLOS ONE